# Evaluation of Sagittal Spinopelvic Alignment on Analgesic Efficacy of Lumbar Epidural Steroid Injection in Geriatric Patients

**DOI:** 10.3390/medicina58101383

**Published:** 2022-10-01

**Authors:** Hee Jung Kim, Min Gi Ban, Miribi Rho, Woohyuk Jeon, Shin Hyung Kim

**Affiliations:** 1Department of Anesthesiology and Pain Medicine, Anesthesia and Pain Research Institute, Yonsei University College of Medicine, Seoul 03722, Korea; 2Department of Anesthesiology and Pain Medicine, Yongin Severance Hospital, Yonsei University College of Medicine, Yongin 16995, Korea; 3Department of Radiology, Research Institute of Radiological Science, Yonsei University College of Medicine, Seoul 03722, Korea

**Keywords:** geriatric patients, epidural steroid injection, muscle degeneration, spinopelvic alignment, pain management, paraspinal muscles

## Abstract

*Background and Objectives*: The aim of this study was to evaluate the impact of sagittal imbalance based on pelvic incidence–lumbar lordosis (PI-LL) mismatch on the analgesic efficacy of epidural steroid injection in geriatric patients. *Materials and Methods*: Patients aged 65 years or older who received lumbar epidural steroid injections under fluoroscopy were enrolled. The cutoff of PI-LL mismatch >20° was used as an indicator of a marked sagittal imbalance. The cross-sectional area of the psoas and paraspinal muscles, as well as the paraspinal fat infiltration grade were measured. A 50% or more decrease in pain score at four weeks after injection was considered as good analgesia. Variables were compared between PI-LL ≤ 20° and >20° groups and multivariate analysis was used to identify factors related to pain relief after injection. *Results*: A total of 237 patients consisting of 150 and 87 patients in the PI-LL ≤ 20° and >20° groups, respectively, were finally analyzed. Female patients, patients with lumbar surgery history, and the smaller cross-sectional area of the psoas muscles were predominantly observed in patients with sagittal imbalance. There was no difference in analgesic outcome after injection according to the PI-LL mismatch (good analgesia 60.0 vs. 60.9%, *p* = 0.889). Multivariate analysis showed that pre-injection opioid use, moderate to severe foraminal stenosis, and high-graded paraspinal fat infiltration were significantly associated with poor analgesia after injection. *Conclusions*: There was no significant correlation between sagittal spinopelvic alignment and pain relief after lumbar epidural steroid injection for geriatric patients.

## 1. Introduction

Degenerative sagittal imbalance results from degenerative structural change of the spine such as disc space narrowing, collapsed vertebral bodies, and/or muscle degeneration in geriatric patients [1,2,3]. An anterior sagittal imbalance following a loss of lumbar lordosis and an increase in pelvis tilt is often observed in patients with chronic low back pain and lumbar degenerative disease [1,2,3]. The pelvic incidence–lumbar lordosis (PI-LL) mismatch based on spinopelvic measurements on X-ray images is one of the common parameters for evaluating sagittal imbalance, and it is strongly correlated with a sagittal vertical axis of the spine [4]. Severe PI-LL mismatch seems to predispose individuals to adjacent segment disease after lumbar fusion and is associated with residual pain symptoms after surgery [5,6,7].

Lumbar paraspinal muscle abnormalities evaluated with magnetic resonance imaging (MRI) are clinically applied to assess spinal sarcopenia and myosteatosis [8]. Degeneration of the paraspinal muscles by way of reduced cross-sectional area and high fat infiltration appeared to be a risk factor for degenerative adult spinal deformity with sagittal imbalance [9]. Epidural steroid injection (ESI) is one of the standard treatments for lower back or radicular pain in the clinical field [10]. However, there is a lack of current studies that have been conducted on the relationship between pre-procedural sagittal imbalance status, including muscle degeneration evaluation with treatment outcomes after ESI.

Therefore, the aim of this retrospective observational study was to investigate the relationship between sagittal spinopelvic imbalance based on PI-LL mismatch and pain relief after ESI in geriatric patients diagnosed with degenerative lumbar disease. Another goal of this study was to identify the relevant clinical factors that affected the pain outcome of ESI in the study population.

## 2. Materials and Methods

### 2.1. Study Population

Approval from the Institutional Review Board was obtained and, in keeping with the policies for a retrospective review, informed consent was not required (No. 4-2022-0811). This manuscript adheres to the applicable STROBE checklists for observational studies. From 2016 to 2017, we enrolled patients 65 years or older with a diagnosis of degenerative lumbar spinal disease who received fluoroscopy-guided lumbar ESI, including interlaminar, transforaminal, and caudal approaches. Exclusion criteria included the following: patients with neurodegenerative and psychiatric disease or cancer; patients without pain score measurement; patients without X-ray/MRI prior to injection; and patients lacking in complete medical records and/or records that were lost in follow-up within 4 weeks of the procedure. The study flow diagram can be found in Figure 1.

### 2.2. Spinopelvic Measurements

The spinopelvic parameters including PI and LL were measured from the pre-procedural lateral X-ray images of the lumbosacral spine, which were taken in the neutral standing position. Following the standard protocol [5,6], angle measurement tools of the picture archiving and communication system (PACS) workstation were typically used (Figure 2). The PI is measured as the angle between a line drawn perpendicular to the upper end plate of S1 at its midpoint and a line drawn from the midpoint of the upper end plate of S1 to the central axis of the femoral head. The LL is the sagittal Cobb angle measured between the upper end plate of the L1 body and the upper end plate of the S1 body. The PI-LL mismatch was calculated as the difference between PI and LL. For the purpose of this study, the patients were divided into two groups: PI-LL ≤ 20° and PI-LL > 20° group. This cutoff of PI-LL > 20° was regarded as an indicator of a marked sagittal spinopelvic imbalance according to the Scoliosis Research Society (SRS)-Schwab classification of adult spinal deformity [11]. A pain physician (H.J.K) evaluated PI and LL at the initial time point and four weeks after the first measurement. The pain physician was blinded to the first results when determining the second results. An independent radiologist (M.R), blinded to all clinical data, assessed all images pertaining to the study. All members involved in the study were blind to results of the other rater.

### 2.3. CSA and Fat Infiltration Measurements

T1-weighted axial images of MRI were used to evaluate the quantitative measurement of psoas major muscles and paraspinal muscles, which envelope the multifidus and erector spinae muscle as one entity. The cross sectional area (CSA), measured by individual manual outlining of each bilateral psoas muscle and paraspinal muscle, was evaluated at the axial image of the L3/4 disc level (Figure 3), with both considered the middle of the lumbar spine, and previous studies contest to this level as having the largest CSA of paraspinal muscles [12,13]. We calculated the total CSA as the sum of the CSAs of all muscles (bilateral psoas muscles and paraspinal muscles) and normalized the value by the square of body height (cm^2^/m^2^). Calculations were performed with Image J software (Version 1.53n, NIH, Bethesda, MD, USA).

A three-grade system based on previous studies was applied to access a semi-quantitative measurement of fatty content of the paraspinal muscles from the same images used to measure CSA (Figure 4) [8,14]. Fat infiltration was graded as follows: Grade 0, normal (0–10% intramuscular fat), Grade 1, mild (10–50% intramuscular fat), and Grade 2, severe (>50% intramuscular fat). Bilateral paraspinal muscles were graded individually and a final grade was determined as the degree of fat infiltration of the paraspinal muscles. The psoas muscles showed negative correlation between fat infiltration and lower back pain and fat infiltration of paraspinal muscles in previous studies; thereby, grading was not performed for psoas muscles [14,15].

A pain physician (H.J.K) performed all measurements of CSA and fat infiltration grading at the initial time point and four weeks after the first measurement. The pain physician was blinded to the first results when determining the second results. An independent radiologist (M.R), blinded to all clinical data, also evaluated the images. Both raters were blinded to the results of the other evaluator. If there was a discord in grading, a third evaluator (S.H.K) was introduced to achieve a consensus between the two initial evaluators.

### 2.4. Fluoroscopy-Guided Lumbar ESI

All ESI procedures, in accordance to the institutional standard protocol, were performed by two operators with similar training using a C-arm fluoroscopy system (ARCADIS Varic 2013; Siemens Medical Solutions, Erlangen, Germany). After correct positioning of the needle tip in the anteroposterior and lateral views, 1–2 mL of contrast media was injected to confirm epidural spread. Then, a lidocaine and 5 mg of dexamethasone (non-particulate water-soluble steroid) was typically administered into the epidural space.

### 2.5. Patient Demographics and Clinical Data Measurements

Review of electronic medical records was performed to collect patient characteristics, factors associated with pain, and clinical factors. Patient characteristics included age, sex, body mass index (BMI), and previous lumbar surgery history. Factors associated with pain included duration of pain, baseline pain score measured by the numeric rating scale (NRS), and pre-opioid analgesic use one month before the procedure. Radiographic features including lumbar herniated disc, grading of stenosis [16,17], spondylolisthesis or fractures were assessed by the final reports of a lumbar MRI performed by a radiologist not involved in this study. An interlaminar, transforaminal, caudal approach for each ESI was clarified for each patient. Also, if a patient opted to receive an operation within one year of ESI, they were considered to represent a transition from conservative care to surgical intervention. We defined good analgesia after ESI as a decrease of 50% or more in pain score without requirements of an increment of analgesic medication at four weeks after the injection.

### 2.6. Statistical Analysis

Data are expressed as mean ± standard deviation (SD) for continuous variables and number (percentages) for categorical variables. For ordinal data and discontinuous variables, we expressed the data as median and interquartile range (IQR). The normality of distribution was examined by the Shapiro–Wilk test to ensure proper statistical treatment. We used the intra-class correlation coefficient to analyze the intra and inter observer reliabilities in relation to the spinopelvic measurements, CSA, and the Cohen’s weighted kappa coefficient for grade of paraspinal fat infiltration. We performed the independent *t*-test, chi-square test, or Fisher’s exact test to analyze demographics, clinical variables, spinopelvic parameters, and muscle degeneration. Continuous variables with non-normal distribution were analyzed with the Mann–Whitney U test. The primary endpoint was the analgesic effects of ESI determined after 4 weeks of the injection between the two groups. The outcome was analyzed with the chi-square test. In regard to our secondary endpoint, significant univariate variables with a *p*-value threshold of 0.2 were entered into a multivariate logistic regression model to assess their impact on pain relief post-ESI. Values were calculated as adjusted odd ratio (aOR) with a 95% confidence interval (CI). Analyses were conducted with the Statistical Package for the Social Sciences, version 25.0 (IBM Corp, Armonk, NY, USA). A *p*-value < 0.05 was considered statistically significant.

## 3. Results

Overall, a total of 237 patients, with 150 patients (63.3%) in the PI-LL ≤ 20° group and 87 patients (36.7%) in the PI-LL > 20° group, were finally analyzed in this study. (Figure 1).

Patient demographics, pre-procedural clinical data, spinopelvic parameters, and psoas/paraspinal muscle degeneration are summarized in Table 1. Patient age and BMI were similar between the two groups. Female patients showed a higher prevalence of sagittal imbalance with PI-LL > 20° than male patients. Also, spine surgery history was more frequently observed in the PI-LL > 20° group. There were no significant differences in pain-related data such as pain duration, pain score, and opioid analgesic use between the groups. Also, prevalence of lumbar spine pathologies based on MRI findings was comparable between the two groups. In the PI-LL > 20° group, CSA of the bilateral psoas muscles was significantly smaller than those of the PI-LL ≤ 20° group, but not the CSA of the bilateral paraspinal muscles. There was no significant difference in the grade of fat infiltration in the paraspinal muscles between the two groups. The approach method of ESI was similar between the two groups.

The reliabilities for all spinopelvic measurements were excellent, with intra-observer ICCs of 0.912 and 0.908 and inter-observer ICCs of 0.901 and 0.921 for the PI and LL, respectively. The intra- and inter-observer reliabilities of total muscle CSAs were excellent for the psoas major (ICC = 0.915, ICC = 0.932, respectively) and paraspinal muscles (ICC = 0.939, ICC = 0.921, respectively). The intra- and inter-observer reliabilities of fat infiltration grading were also excellent for the paraspinal muscles (kappa = 0.903, kappa = 0.868, respectively).

At four weeks after ESI, there was no difference in the proportion of patients with good analgesia according to the PI-LL mismatch before injection (Table 2). Also, the transition rate to spine surgery within one year after ESI was similar between the two groups.

Univariate and multivariate analyses for factors investigated against poor pain relief after ESI can be found in Table 3. The spinopelvic parameters were not associated with pain outcome after ESI. In multivariate analysis, opioid usage before ESI (aOR = 2.021, 95% CI = 1.127–3.622, *p* = 0.018), moderate- to severe-graded foraminal stenosis (aOR = 2.672, 95% CI = 1.274–5.603, *p* = 0.009), and high-graded paraspinal fat infiltration (aOR = 5.021, 95% CI = 2.057–12.258, *p* < 0.001) were identified as independent factors of poor analgesic pain relief after ESI (Table 3). There was no significant association between age, sex, pain duration, or presence of spondylolisthesis and pain outcome after ESI after adjustment of other variables.

## 4. Discussion

Degenerative sagittal imbalance is associated with back pain and poor quality of life in the geriatric population [18]. The spinopelvic malalignment based on PI-LL mismatch predisposed adjacent segment disease after lumbar fusion and was related to a higher risk of revision surgery [5]. This PI-LL mismatch affected post-operative residual symptoms, such as low back or radicular pain and numbness in patients undergoing short-segment lumbar fusion surgery [6]. High pelvic incidence (PI) increases the risk of sagittal imbalance after spine fusion and is a predictive factor for degenerative spondylolisthesis [7]. For these reasons, we hypothesized that there would be an association between severe PI-LL mismatch and poor analgesic outcome in geriatric patients. However, our study showed that sagittal spinopelvic alignment did not influence pain outcome after lumbar epidural steroid injection. In the present study, PI-LL mismatch greater than 20° was more frequent in female than male patients and patients with a lumbar spine surgery history. Also, a smaller CSA of the psoas muscles was observed in patients with a marked PI-LL mismatch. Psoas major and paraspinal muscles play an important role in the segmental stability of the lumbar spine, and these muscle groups are directly linked to the motor control of the pelvis. A previous study showed a stronger correlation between whole body skeletal muscle mass and CSA of the psoas muscles than those of the paraspinal muscles, and CSA of the psoas muscles was negatively correlated with pelvic tilt in degenerative lumbar spinal stenosis [19]. In contrast, CSA of the multifidus muscle, but not of the psoas muscles, was smaller in female kyphosis patients with scoliosis who scheduled correction surgery than in controls [20]. Therefore, although there were some different results according to spine pathologies and related complex compensation mechanisms, diminished lumbar muscle mass seemed to be associated with sagittal imbalance of the spine. However, this sagittal imbalance with a loss of localized lumbar muscle mass was not associated with impaired handgrip strength, which may indicate global skeletal muscle weakness [12].

Spinopelvic parameters were correlated with lumbar muscle volumes but not with muscle fat infiltration in asymptomatic young adults [21]. Our study showed higher fatty content of paraspinal muscles suggesting that poor muscle quality might be a more significant factor associated with pain relief after ESI in geriatric patients, rather than PI-LL mismatch with reduced CSA of the lumbar muscles. Considering these results, muscle size alone without taking into account composition or function, may not be the ideal representation of muscle degeneration of the spine. The degree of fatty infiltration of the paraspinal muscle may be a more important factor in regard to severe lower back pain and disability [8]. A recent report demonstrated a favorable pain outcome after lumbar surgery for patients with lower fat infiltration of the paraspinal muscles and low association of CSA, which is congruent to our study [22]. High paraspinal fat infiltration had a significant correlation with poor quality of life in older patients with degenerative lumbar disease, which showed stronger correlation with poor functional status than the sagittal imbalance [23]. Thus, the relationship between health-related quality of life and sagittal imbalance can depend on the quality of lumbar muscle. Also, a lower level of physical activity was closely related to higher fat content of the paraspinal muscles [24]. This suggests that paraspinal fat infiltration may be an indicator as to whether a patient will be active in physical exercise after injections.

In the current study, the results revealed an association between the severity of foraminal stenosis and analgesic outcome of ESI. A result of the previous study indicated that although transforaminal ESI was effective in pain relief without regard to foraminal stenosis severity, patients with severe foraminal stenosis reported a significantly smaller drop in pain scores as time passed [16]. Similar to previous results, our study indicated that a younger age may correlate with superior pain relief [25]. However, age was not an independent factor associated with pain relief after ESI in our study population. A previous report showed that an increase of pre-injection opioid dosage was associated with decrease of improvement in disability at 3 months, but no difference at 12 months after cervical and lumbar ESIs [26]. In our study, there was a correlation between pre-injection opioid use and analgesic outcome in the short term. Further studies are warranted to investigate dose-dependent or chronic pre-opioid use against long-term analgesic efficacy of ESI.

There are several limitations in this study. There may be information bias due to the retrospective nature of the current study. Also, this study was performed in a single-center institution with a small sample size involving a homogenous racial study population. Due to the lack of whole spine X-rays available for a majority of patients, PI and LL were measured only as spinopelvic parameters for assessment of sagittal imbalance. Although previous studies support a strong association of muscle CSA and paraspinal fat infiltration at the L3-L4 level with other levels [8,24,27], a cross-sectional evaluation at a single level may not properly reflect an entire muscle mass and composition [28]. Lastly, the current results are a representation of a cross-sectional relationship between spinopelvic parameters or lumbar muscle degeneration and the analgesic efficacy of ESI. Further studies, including longitudinal studies, are needed to investigate the causal relationship within variables, in particular the global sagittal alignment parameters in relation to muscle morphology, function, and strength.

## 5. Conclusions

In conclusion, sagittal imbalance based on spinopelvic measurement was not associated with pain outcome of ESI. Instead, there was a significant association between high paraspinal fat infiltration and poor pain relief after ESI. Therefore, the analgesic efficacy of ESI has a higher association to the quality of paraspinal muscles than sagittal imbalance, which might be linked to the loss of muscle mass in geriatric patients with symptomatic degenerative lumbar spinal disease.

## Figures and Tables

**Figure 1 medicina-58-01383-f001:**
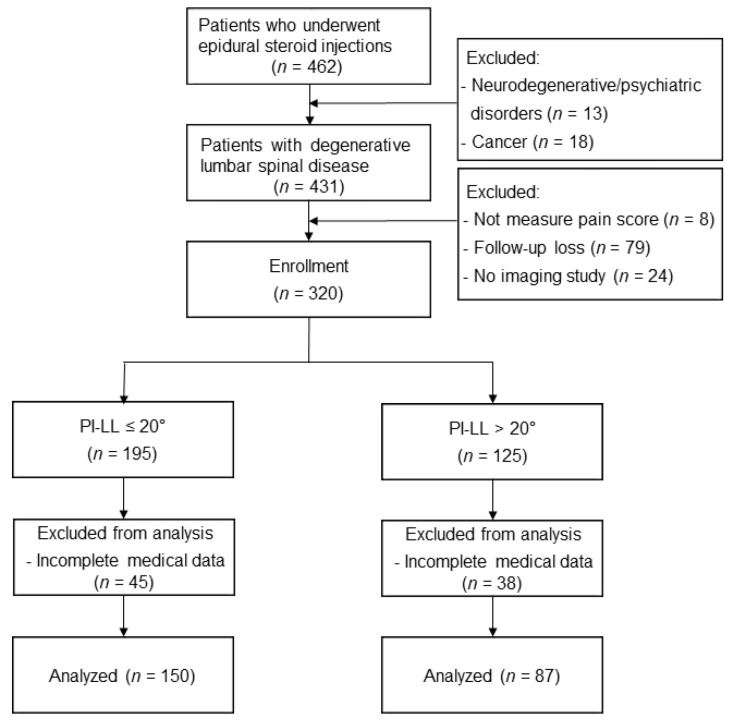
Flow diagram of the study. PI, pelvic incidence; LL, lumbar lordosis.

**Figure 2 medicina-58-01383-f002:**
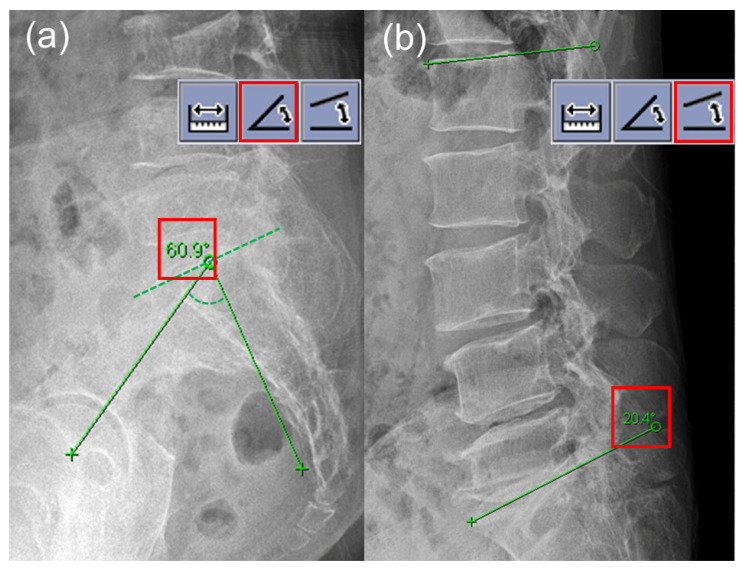
Measurement of pelvic incidence (**a**) and lumbar lordosis (**b**) on a lateral X-ray image of the lumbosacral spine.

**Figure 3 medicina-58-01383-f003:**
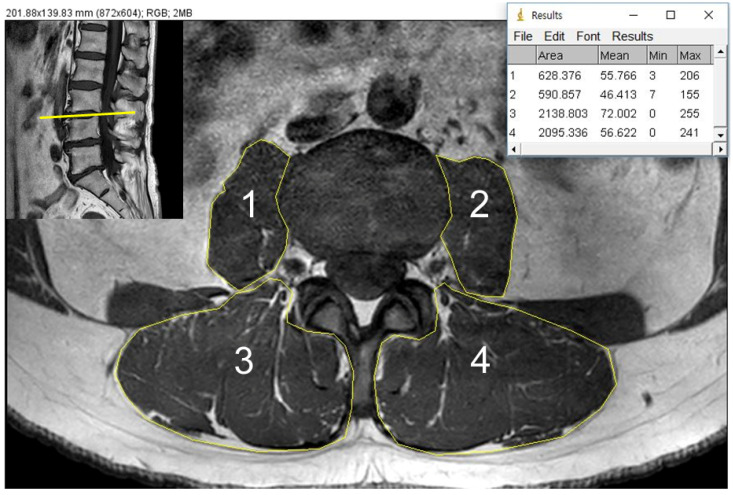
Method to measure cross sectional area bilateral psoas major (1, 2) and paraspinal muscles (3, 4) shown at L3/4 disc level on magnetic resonance image.

**Figure 4 medicina-58-01383-f004:**
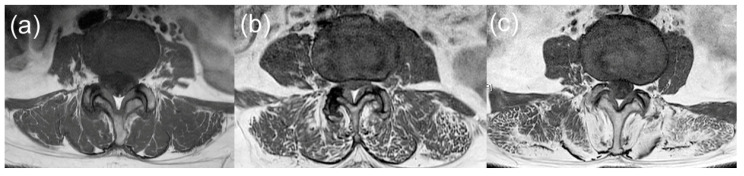
Intramuscular fat infiltration classification system (L3/4 disc level shown on magnetic resonance image): (**a**) Grade 0: normal (0–10% fat infiltration), (**b**) Grade 1: mild (10–50% fat infiltration), and (**c**) Grade 2: severe (>50% fat infiltration).

**Table 1 medicina-58-01383-t001:** Comparison of patient characteristics, clinical data, spinopelvic parameters, and muscle degeneration according to the pre-procedural pelvic incidence–lumbar lordosis (PI-LL) mismatch.

Variables	PI-LL ≤ 20° (*n* = 150)	PI-LL > 20° (*n* = 87)	*p*-Value
Patient characteristics			
Age, years	73.68 ± 5.52	73.48 ± 6.58	0.814
Sex, M/F	62 (41.3)/88 (58.7)	19 (21.8)/68 (78.2)	0.002
Body mass index, kg/m^2^	24.45 (22.61–26.39)	24.55 (22.52–27.10)	0.899
Spine surgery history, *n*	29 (19.3)	31 (35.6)	0.005
Pain-related data			
Pain duration, months	6.00 (1.00–18.75)	7.00 (2.00–36.00)	0.227
Baseline pain score, NRS 0–10	7.13 ± 1.75	7.11 ± 1.59	0.936
Opioid use before injection, *n*	44 (29.3)	34 (39.1)	0.124
Pre-procedural MRI findings, *n*			
Herniated disc	146 (97.3)	85 (97.7)	0.862
Foraminal stenosis			0.572
No to mild	34 (22.7)	17 (19.5)	
Moderate to severe	116 (77.3)	70 (80.5)	
Central stenosis			0.816
No to mild	110 (73.3)	65 (74.7)	
Moderate to severe	40 (26.7)	22 (25.3)	
Compression fracture	22 (14.7)	18 (20.7)	0.233
Spondylolisthesis	34 (22.7)	14 (16.1)	0.225
Spinopelvic parameters			
PI	46.50 ± 11.36	58.06 ± 10.30	<0.001
LL	38.75 ± 10.64	29.28 ± 10.69	<0.001
PI-LL	7.75 ± 9.16	28.78 ± 6.83	<0.001
Muscle degeneration			
Cross-sectional area, cm^2^/m^2^			
Psoas muscles	5.40 ± 1.61	4.76 ± 1.39	0.001
Paraspinal muscles	18.61 ± 3.04	18.03 ± 3.25	0.163
Psoas + paraspinal muscles	24.01 ± 3.70	22.78 ± 3.84	0.016
Paraspinal fat infiltration grade			0.238
Grade 0, normal	32 (21.3)	10 (11.5)	
Grade 1, mild	73 (48.7)	50 (57.5)	
Grade 2, severe	45 (30.0)	27 (31.0)	
Epidural approaches, *n*			0.235
Interlaminar	5 (3.3)	1 (1.1)	
Transforaminal	132 (88.0)	73 (83.9)	
Caudal	13 (8.7)	13 (14.9)	

Values are mean ± SD, median (interquartile range) or number of patients (%). NRS, numeric rating scale; MRI, magnetic resonance imaging.

**Table 2 medicina-58-01383-t002:** Comparison of treatment outcome after epidural steroid injections according to the pre-procedural pelvic incidence–lumbar lordosis (PI-LL) mismatch.

	PI-LL ≤ 20° (*n* = 150)	PI-LL > 20° (*n* = 87)	*p*-Value
4 weeks follow-up, *n*			0.889
Good analgesia	90 (60.0)	53 (60.9)	
Poor analgesia	60 (40.0)	34 (39.1)	
1 year follow-up, *n*			0.421
Transition to spine surgery	17 (11.3)	13 (14.9)	

Values are number of patients (%). Good analgesia was defined as a 50% or more decrease in pain score without increasing analgesic dosage at four weeks after injection.

**Table 3 medicina-58-01383-t003:** Univariate and multivariate analysis: Associated factors of poor pain relief after epidural steroid injection.

	Crude OR	95% CI	*p*-Value	Adjusted OR	95% CI	*p*-Value
Patient characteristics						
Age, ≥ 75 years	1.742	1.029–2.950	0.039	1.278	0.707–2.308	0.417
Female	1.506	0.860–2.637	0.152	0.931	0.478–1.812	0.833
BMI, ≥ 25 kg/m^2^	1.134	0.669–1.920	0.641			
Spine surgery history, yes	1.118	0.616–2.028	0.714			
Pain-related data						
Pain duration, ≥ 1 years	1.514	0.893–2.565	0.123	1.461	0.824–2.589	0.194
Pain score, ≥ 7 on NRS	1.155	0.672–1.985	0.601			
Opioid usage, yes	1.891	1.090–3.281	0.023	2.021	1.127–3.622	0.018
Pre-procedural MRI findings						
Herniated disc, yes	0.650	0.128–3.291	0.603			
Central stenosis, yes	1.360	0.757–2.444	0.304			
Foraminal stenosis, yes	2.255	1.127–4.510	0.022	2.672	1.274–5.603	0.009
Compression fracture, yes	1.017	0.508–2.036	0.962			
Spondylolisthesis, yes	1.700	0.898–3.218	0.103	1.447	0.730–2.869	0.289
Spinopelvic parameters						
PI	1.008	0.987–1.030	0.465			
LL	1.003	0.981–1.026	0.765			
PI-LL	1.004	0.985–1.024	0.674			
PI-LL mismatch >20°, yes	0.962	0.560–1.652	0.889			
Muscle degeneration						
Cross-sectional area, cm^2^/m^2^						
Psoas muscles	0.902	0.761–1.070	0.237			
Paraspinal muscles	1.026	0.944–1.115	0.549			
Psoas + paraspinal muscles	1.001	0.934–1.071	0.995			
Paraspinal fat infiltration grade						
Grade 0, normal	1.000			1.000		
Grade 1, mild	2.042	0.896–4.656	0.089	2.032	0.876–4.713	0.098
Grade 2, severe	4.849	2.027–11.601	<0.001	5.021	2.057–12.258	<0.001
Epidural approach						
Caudal	1.000					
Interlaminar	0.500	0.078–3.223	0.466			
Transforaminal	0.627	0.277–1.422	0.264			

OR, odds ratio; CI, confidence interval; BMI, body mass index; NRS, numeric rating scale; PI, pelvic incidence; LL, lumbar lordosis.

## Data Availability

Data are available upon request to corresponding author.

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
