# Peer review of "Evaluation of Sagittal Spinopelvic Alignment on Analgesic Efficacy of Lumbar Epidural Steroid Injection in Geriatric Patients"

_medicina, 2022, doi:10.3390/medicina58101383_

Round 1

Reviewer 1 Report

Thank you for the excellent study.

It is interesting and novel.

Please can you describe -is the corticosteride water soluble and why only 5 mg ?

which kind of local anaesthesia?

Author Response

Reviewer 1

Thank you for the excellent study.

It is interesting and novel.

Please can you describe -is the corticosteride water soluble and why only 5 mg ?

which kind of local anaesthesia?

Response)

  • We truly thank you for your thorough review and constructive comments on our study.
  • Dexamethasone does not form particles or aggregates large enough to cause an embolism, based on published case reports of paraplegia, quadriplegia, or stroke following ESI (Pain Med 2008; 9: 227-34.). In 2020, Donohue et al. reported that there was no significant difference in pain relief at any point between nonparticulate and particulate steroids and recommended the use of nonparticulate corticosteroids in ESI given the safety concerns associated with particulate corticosteroids (Korean J Pain 2020; 33: 192-8.). Considering the potential risk of catastrophic complications, we have used nonparticulate steroid preparations as first-line agents when performing ESI.
  • There was no standard dose of dexamethasone for ESI. The World Institute of Pain (WIP) Benelux working group recommended that the number of ESIs should be adjusted according to the clinical response, suggesting that a 2-week interval for additional ESI may be appropriate for proper evaluation and minimization of endocrine side effects, and the lowest effective dose should be used for ESI (10 mg for dexamethasone phosphate) (Pain Pract 2019; 19: 61-92.). We have typically used 5mg of dexamethasone for elderly patients because this population may be more vulnerable to potential systemic side effects of steroid use.
  • A mixture of dexamethasone and ropivacaine induced a pH-dependent crystallization in vitro, thus, we have typically used a ‘lidocaine’ according to the institutional protocol for ESI.
  • We have added this information in the revised manuscript briefly.

Reviewer 2 Report

Evaluation of sagittal spinopelvic alignment on the analgesic efficacy of lumbar epidural steroid injection in geriatric patients.

A brief summary

      The aim of this study was to evaluate the impact of sagittal imbalance based on pelvic incidence-lumbar lordosis (PI-LL) mismatch on the analgesic efficacy of epidural steroid injection in geriatric patients. The main contribution of this study is that more likely diminished lumbar muscle mass with higher fat content than sagittal spinopelvic alignment seems to be associated with the sagittal imbalance of the spine and bad pain outcomes and poor quality of life and that poor muscle quality might be a more significant factor associated with pain relief after epidural steroid injection in geriatric patients.

      This study shows that paraspinal fat infiltration has a significant correlation with poor quality of life and poor functional status and may be an indicator as to whether a patient will be active in physical exercise after injections or not.

General concept comments

    This study is retrospective observational, the review topic is highly relevant because low back pain is one of the most common health problems among older adults, resulting in chronic pain and disability, the references are appropriate and actual and the results of this study make the diagnosis of possible treatment failure more precise. Approval from the Institutional Review Board was obtained for a retrospective review, informed consent was not required.

    The title of the scientific paper contains a brief description of the content. The title should accurately describe the content of the article.

    The manuscript is clear and presented in a well-structured manner, the experimental design is appropriate to test the hypothesis, and the tables and figures properly show the data that are easy to interpret and understand.

    The scientific content is sharing the author's original research work, which could be important, valid, and relevant to other scientists in the same field, the conclusions are consistent with the evidence and arguments presented. Although they were similar reviews published recently, this current review specifies the possibility of diagnosis and predicts the response to lumbar epidural steroid injection in geriatric patients.

    The methods section provides sufficient details for other scientists to reproduce the experiments presented in the paper.

    There were published 13 from 28 publications from the cited references within the last 5 years, and they are relevant. There is one self-citation.

    Statistical analysis: Analytical methods are appropriate for the type of data, data are expressed as mean ± standard deviation (SD) for continuous variables and percentages for categorical variables. A P-value < 0.05 was considered statistically significant. The analyses are performed with technical standards and the data are robust enough to draw conclusions. The raw data are available upon request to the corresponding author.

Summary

Novelty: The results provide an advancement of the current knowledge.

Scope: The work fits the journal scope

Significance: The results are significant.

Quality: Good.

Scientific Soundness: The study is applicable.

Interest to the Readers: Yes

Overall Merit: The work advance the current knowledge of low back pain in geriatric patients

English Level: Good.

Author Response

Reviewer2

Evaluation of sagittal spinopelvic alignment on the analgesic efficacy of lumbar epidural steroid injection in geriatric patients.

A brief summary

  • The aim of this study was to evaluate the impact of sagittal imbalance based on pelvic incidence-lumbar lordosis (PI-LL) mismatch on the analgesic efficacy of epidural steroid injection in geriatric patients. The main contribution of this study is that more likely diminished lumbar muscle mass with higher fat content than sagittal spinopelvic alignment seems to be associated with the sagittal imbalance of the spine and bad pain outcomes and poor quality of life and that poor muscle quality might be a more significant factor associated with pain relief after epidural steroid injection in geriatric patients.
  • This study shows that paraspinal fat infiltration has a significant correlation with poor quality of life and poor functional status and may be an indicator as to whether a patient will be active in physical exercise after injections or not.

General concept comments

  • This study is retrospective observational, the review topic is highly relevant because low back pain is one of the most common health problems among older adults, resulting in chronic pain and disability, the references are appropriate and actual and the results of this study make the diagnosis of possible treatment failure more precise. Approval from the Institutional Review Board was obtained for a retrospective review, informed consent was not required.
  • The title of the scientific paper contains a brief description of the content. The title should accurately describe the content of the article.
  • The manuscript is clear and presented in a well-structured manner, the experimental design is appropriate to test the hypothesis, and the tables and figures properly show the data that are easy to interpret and understand.
  • The scientific content is sharing the author's original research work, which could be important, valid, and relevant to other scientists in the same field, the conclusions are consistent with the evidence and arguments presented. Although they were similar reviews published recently, this current review specifies the possibility of diagnosis and predicts the response to lumbar epidural steroid injection in geriatric patients.
  • The methods section provides sufficient details for other scientists to reproduce the experiments presented in the paper.
  • There were published 13 from 28 publications from the cited references within the last 5 years, and they are relevant. There is one self-citation.
  • Statistical analysis: Analytical methods are appropriate for the type of data, data are expressed as mean ± standard deviation (SD) for continuous variables and percentages for categorical variables. A P-value < 0.05 was considered statistically significant. The analyses are performed with technical standards and the data are robust enough to draw conclusions. The raw data are available upon request to the corresponding author.

Response)

  • We truly thank you for your thorough review and constructive comments on our study.